# A matter of principle or a matter of money? How fairness evaluations change with experimental currencies

Marina Chugunova[1], Wolfgang J. Luhan[2]*

1 Max Planck Institute for Innovation and Competition, Munich, Bavaria, Germany, 2 Faculty of Business and Law, University of Portsmouth, Portsmouth, Hampshire, United Kingdom

* wolfgang.luhan@port.ac.uk

## Abstract

We investigate how the information on monetary outcomes influences perceptions of fairness of income redistributions. In an economic experiment, participants initially rated the fairness of a redistribution scheme without knowing the exchange rate of experimental tokens to real money. After learning the monetary value of tokens, the participants adjusted their fairness ratings, generally perceiving redistributions that generate higher income for themselves as fairer. As the redistribution itself did not change, our findings suggest that awareness of monetary consequences affects perceptions of redistribution beyond mere self-interest.

## Introduction

Incentives are a cornerstone of behavioral economics [1], but the way incentives are introduced can add a frame to the decision in question and lead to behavioral changes [2,3]. Behavioral economics, using controlled experimental settings, has made a major contribution to our understanding of social preferences, fairness and their influence on economic behavior [4–7]. Significant attention has been dedicated to the redistributive principles which people apply when judging fairness [8,9]. In this study, we consider whether the perception of fairness is a matter of principle only or also a matter of money. In particular, we examine how the perception of the same redistribution decision changes when the outcome is presented in experimental currency units of uncertain value versus when it is explicitly stated in monetary terms.

To answer this question, we conduct an online experiment with a within-subject design: We compare fairness evaluations of income redistribution when expressed in uncertain token values versus real currency. Participants earn tokens of different types through luck, effort, and talent-related tasks. They are matched into pairs and their tokens are subsequently redistributed by a third party. Participants know that each token type has a value from a certain range and that these values will determine the payment at the end of the study. Initially, participants rate the fairness of the redistribution in tokens. At the end of the study, we reveal the value of each token

**Data availability statement:** The data set underlying this study may be found at https://doi.org/10.17605/OSF.IO/JUCEV.

**Funding:** This study was financially supported by the Nuffield Foundation in the form of a grant awarded to WL (FR-000000326). This study received funding from the Deutsche Forschungsgemeinschaft (DFG) through the Collaborative Research Center Transregio TRR 190 (project number 280092119), to which MC is affiliated. The funders had no role in study design, data collection and analysis, decision to publish, or preparation of the manuscript.

**Competing interests:** The authors have declared that no competing interests exist.

and allow participants to rate the fairness of the redistribution again. Both ratings are hypothetical and do not affect the earnings of the participants. As the value of the tokens is the same for everybody, their relative earnings compared to their counterpart remain unaffected. Therefore, if the fairness of redistribution is defined by normative principles (i.e., what should or should not be redistributed), introducing the monetary value of tokens should not affect the perception of fairness, and most fairness theories predict that perceptions of fairness will not change.

We document that expressing incentives in their monetary terms can serve as a frame and reinforce the role of self-interest in evaluating the fairness of redistributions: Participants rate the same distributions differently depending on whether they know the monetary value of the experimental currency and adjust their fairness ratings, often reflecting a shift toward more self-interested evaluations. Our results highlight the important role of self-serving biases and motivated reasoning [10] for experimental designs, showing that even small changes can significantly impact people's perception of a situation and their behavior.

Our study connects and builds on several strands of literature: crowding out of social preferences by monetary incentives, perceptions and evaluations of fairness and, relatedly, the role of self-serving bias for those evaluations.

Crowding out of social preferences by monetary incentives has been discussed extensively in the literature (see, for an overview [11]), starting with the prominent field study by Titmuss [12]. The argument is that the prospect of payments shifts the focus from other motivations [13] to purely monetary considerations, triggering a (simplified) cost-benefit analysis [14]. Had there been a social aspect before, it is now replaced by the monetary value of the choice [2].

In the context of economic experiments, this leads to the debate whether monetary incentives are necessary (see, e.g., [1,15,16]) or even counter-productive in certain areas [17], and the way these should be implemented [18]. Extending this topic, there is a significant body of literature on the effects of varying stake sizes on observed behavior in experiments (e.g., [19]). Concerning social preferences, in particular fairness considerations, there appears to be no effect on transfers in dictator games [20–22], or proposed shares in the ultimatum game, but a decreasing acceptance threshold of the second-movers in this game points towards a shift in fairness preference with increasing stake sizes [22–24]. We extend this research by introducing an aspect of uncertainty about the monetary value of the currency used in the experiment. Our question is whether participants judge the same income distribution differently before and after learning the exchange rate of the experimental currency to cash.

By introducing uncertainty to monetary value, our study may resemble Rawls' [25] 'veil of ignorance' describing that people would opt for a just society if they did not know their final position in society, or Chen and Zhong [26], who show that additional uncertainty can increase moral behavior. Yet, the crucial difference is that participants know their and other participants' earnings—so they can fully judge their relative position—but they do not know what this will translate to in monetary terms. That is, participants know their earnings in experimental currency units but do not know the exact exchange rate. One could argue that this could create a "hot vs.

cold decision" shift, as uncertainty about monetary values could decrease emotional responses, thus shifting the reported perception of distributions [27]. Brandts and Charness [28], however, do not find a difference between hot and cold decisions, so the perceived fairness should not change.

Fairness principles are a focus of active and dynamic research (e.g., [8,9]). It has been documented that people might adjust their fairness evaluations once they themselves are directly affected by the decision as compared to when they are an uninvolved third party with no stake in the outcome. For example, in Luhan et al. [29], participants evaluate redistributions after a real-effort task as an external observer. Redistributing income that was randomly allocated is generally seen as fair, while income that required work should be left with the person who put in the effort. This changes drastically when the same participants take the role of the working party: high-earners find it fair not to redistribute at all, while low-earners find egalitarian redistributions most fair. This moral hypocrisy, where moral judgments and stated fairness principles are adjusted to the personal earnings (e.g., [30,31]) can, in part, be explained by the real costs of social image concerns. Evaluations follow social norms and socially accepted principles if there are no costs (for the person making the judgment) involved. However, the introduction of real, often monetary, costs leads to a different evaluation. A self-serving bias will not completely crowd out fairness beliefs and principles but significantly alter the evaluation in the decision maker's favor [32]. In our study, the presence of the self-serving bias as such remains constant – participants are always a party affected by the decision. However, expressing the redistribution in experimental currency units with uncertain value as opposed to their monetary equivalent appears to reinforce self-interest.

In our study, we narrow down the role that a self-serving bias or crowding out of social motivations by monetary interests can play in evaluations of redistributive decisions. Participants know that the situation they are evaluating concerns themselves, what they have earned and even their status relative to other participants. The only aspect that changes between the two evaluations is the knowledge of the monetary value of their tokens. We contribute to the literature by demonstrating that knowledge of monetary value can serve as a frame. It intensifies self-serving bias and influences fairness evaluations in redistributive decisions, even when individuals are fully informed about their relative performance and status.

Our study offers subtle but valuable insights into how the presentation of incentives can affect social judgments, making a primarily methodological contribution. However, our results also help explain why perceptions of redistributive decisions may differ when discussed in abstract terms (e.g., during compensation negotiations or in political decision-making) versus when implemented with real outcomes.

## Materials and methods

This study was conducted in conjunction with a companion project reported in Chugunova and Luhan [33]. Fig 1 displays the outline of the experiment. Our analysis focuses on the comparison of evaluations submitted in the last two stages of the study, marked in yellow.

We conducted an online experiment with a within-subject design. Participants first generated their endowment: They earned tokens by completing three tasks, each representing a key determinant of income central to distributional fairness principles: luck, effort, and talent [8]. Participants could earn 0 or 100 tokens via a coin toss (luck); they counted the zeros in two matrices of zeros and ones for 100 tokens each (effort); finally, they could earn 100 tokens for solving one of Raven's Progressive Matrices (talent) [34].

Each task created earnings in distinct tokens, the exact value of which was not disclosed until the end of the experiment. Participants were informed that the exchange rate could range from 1 to 6 cents for each token type. The separate values of tokens allow us to distinguish between four fairness principles of distribution (see, e.g., [4,8,9,29,35]): egalitarian, choice egalitarian, meritocratic, and libertarian. Similar tasks are widely used in the literature (see [36]) and Luhan et al. [29] show that they effectively capture production based on luck, effort, and talent. Participants typically view the coin

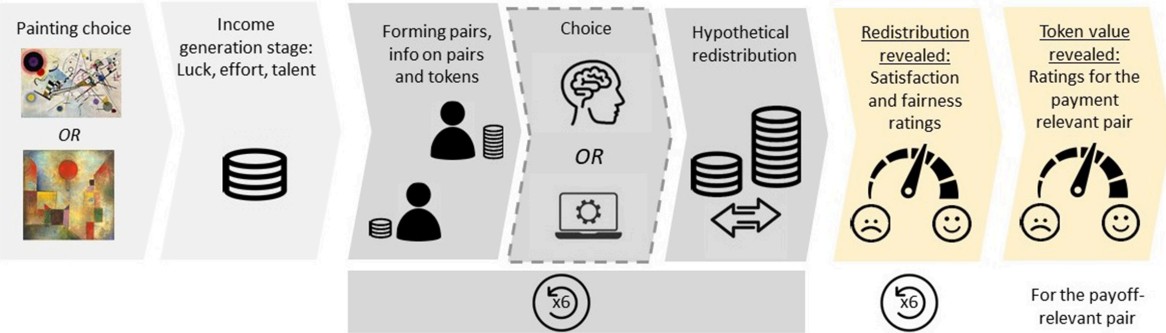

**Fig 1**. **Outline of the experiment.** *Notes:* Stage 4, depicted with a dashed line, introduced between-subject variation analyzed in Chugunova and Luhan [33]. The current paper focuses on the within-subject comparison of evaluations submitted in the final two stages of the study, marked in yellow.

toss as luck-based, zero-counting as effort-based, and an intelligence test as primarily talent-based with a minor effort component. We therefore consider these tasks suitable for analysing Konow's [8] fairness principles.

The fact that the monetary value of the tokens was initially unknown and could vary for each category forced participants to view all tasks as equally important and not to focus on single income elements or the total number of tokens. This uncertainty in the value of each token type allows us to consider how monetary equivalents can change the fairness perception and potentially exacerbate self-interest.

After earning the tokens of different types, participants were randomly matched into pairs 6 times (6 periods with a perfect stranger matching). Each time, they learned the token portfolio of the matched participant and their own. They knew that the tokens might be redistributed within the pair. For simplicity and conceptual clarity, we did not allow for continuous redistribution: The tokens could have been redistributed evenly, transferred to one of the parties, or remained untouched. The outcome of one randomly drawn pair determined the participants' payments.

For each of the six pairs, participants saw how many tokens of each type they and the matched participant earned and reported what redistribution they would find fair for this pair. This report was hypothetical and did not affect the actual distribution, but purely reflected their fairness preferences, social acceptability, and self-interest.

In the following stage, they saw all six pairs sequentially again and learned how the tokens were redistributed within each pair by a third party. For each pair, they answered how fair they found the redistribution, and how satisfied they were with it. Both measures were reported on 7-point Likert scales and had no effect on participants payoffs (see S1 Fig and S2 Fig for the snippets of the screens). One could argue that fairness and satisfaction should be directly linked, but Charness and Grosskopf [37] do not find a strong correlation between distributional fairness and happiness, so we measured these concepts separately. This report was hypothetical. Next, participants learned which period determined their payment and the value of each token type. Each earned token was worth 4 cents in the luck task, 2 cents in the effort task, and 3 cents in the talent task for all participants.

Finally, for the payoff-relevant pair, participants reported *again* how fair they found the redistribution, this time in real currency, and how satisfied they were. This report–again–was hypothetical and did not affect their earnings. This setting allows us to record several distinct evaluations of the payoff-relevant period: the redistribution the participant finds fair, and their fairness and satisfaction ratings in tokens and in money. We, therefore, can consider how fair the decision was viewed under uncertainty about the value of the tokens and how the fairness perception changed once the value of the token was revealed. As the distribution decision remained the same and the outcomes of the players relative to each other did not change, we can focus on the role of revealing the monetary value. In this environment, if fairness evaluations

are based solely on principles of what should or should not be redistributed for each category of earnings (e.g., [4]), then expressing the same redistribution in tokens or in money should not change the evaluation.

The experiment concluded with a questionnaire that contained some basic demographic characteristics, self-evaluations of trust, risk, social justice orientation scale [38], and several other questions related to the companion study.

**Procedures.** The project received ethical approval from the Faculty Ethics Committee (Business and Law) of the University of Portsmouth (BAL/2020/31/LUHAN) and the Ethics Board of the WiSo Laboratory at the University of Hamburg. The experiment was implemented online using oTree [39] with participants recruited from the subject pool of the WiSo Laboratory of the University of Hamburg using hroot [40]. We used procedures typical for laboratory experiments such as simultaneous participation in 'session groups' and the option to ask clarification questions. In total, 334 participants took part in the experiment. The sessions were gender-balanced. The average age of participants was 25.4. 98% of participants were students. The average payment was 9.06 Euro for 45 minutes. The data collection took place between 30 November and 18 December 2020. Participants' written and informed consent was obtained prior to the experiment. The data are available at OSF: https://osf.io/jucev/

## Results

Our analysis focuses only on the payoff-relevant period, for which we have separate evaluations before and after learning the token values. We elicited separate ratings of fairness and satisfaction. Our data confirm that the two concepts are interrelated (Pearson's correlation of 0.73 for the ratings under uncertainty, $p<0.0001$) but are not identical. Participants can, e.g., be satisfied with getting more tokens, yet they would not consider this fair.

Analyzing the fairness evaluation of payoff distributions, we find a significant change when participants learned the value of tokens. The average distribution under uncertainty was evaluated as "somewhat fair" (0.91 on a scale from –3 to 3, where –3 corresponds to "very unfair" and 3 to "very fair"). Column 1 of Table 1 reports the results of an OLS regression that considers the evaluation of fairness with known monetary equivalent (*Adjusted Fairness*) controlling for the initial evaluation of fairness under uncertainty (*Initial Fairness*), the participant's earnings (*Final Payoff*), the payoff the participant would have received if there was no redistribution (*Original Payoff* ) and the payoff they would receive if their preferred hypothetical redistribution for the pair was implemented (*Preferred Payoff*). Further, we control for the payoff of the other player in the pair (*Partner Payoff*). All payoff measures are in cents. The two alternative payoff outcomes might serve as counterfactuals for evaluating the fairness of the realized redistribution. We control for basic demographic features such as gender and age and for the features of the experimental environment that differed between subjects as described in Chugunova and Luhan [33]. The final evaluation is strongly driven by the initial fairness evaluation. The regression results suggest a self-serving bias: if the final payoff is higher, the ranking of fairness also significantly increases. Participants revise their evaluation upon learning the value of their tokens. High earners re-evaluate the situation as fairer, while low earners perceive it as less fair than when they saw the same distributions in tokens instead of money.

In 140 cases (41.9%), participants received exactly the distribution they had deemed fair when knowing only the token amounts. These participants rate the distribution as "fair", on average (2.2 on the scale from –3 to 3). In the regression analysis, we find a significant but negative effect of learning the monetary values (column 2 in Table 1). This coefficient is significant only at the 5% level and is not robust to estimation using an ordered logit model (S1 Table). We, therefore, interpret it as evidence that the role of the monetary value is less important when the redistribution corresponds exactly to what the participant would deem fair. Potentially, participants may even decrease their fairness rankings when they receive exactly what they deemed fair and happened to earn more.

In 142 cases (42.5%), the original distribution of tokens remained unchanged. Under uncertainty, when distribution of tokens remained unchanged, participants ranked it as "somewhat fair" (1.31 on the scale between –3 and 3). When any tokens were redistributed, participants, on average, rated the outcome as less fair (0.6 on the scale between –3 and 3).

                                     

**Table 1**. OLS regression: Adjustment of fairness and satisfaction rankings after learning the token value.

| | (1) Adjusted Fairness | (2) Adj. Fairness: Realized = Preferred | (3) Adj. Fairness: Redistributed | (4) Adj. Fairness: Not Redistributed | (5) Adjusted Satisfaction |
|---|---|---|---|---|---|
| Initial Fairness | 0.8104*** (0.0340) | 0.6643*** (0.1014) | 0.8072*** (0.0438) | 0.7799*** (0.0561) | |
| Initial Satisfaction | | | | | 0.6447*** (0.0526) |
| Final Payoff (in cents) | 0.0011** (0.0004) | −0.0009* (0.0005) | 0.0011* (0.0005) | 0.0012*** (0.0003) | 0.0027*** (0.0005) |
| Original Payoff (in cents) | −0.0003 (0.0003) | 0.0008 (0.0004) | −0.0005 (0.0004) | | −0.0008* (0.0004) |
| Preferred Distribution (in cents) | −0.0005 (0.0003) | | 0.0001 (0.0004) | −0.0011*** (0.0003) | −0.0009** (0.0003) |
| Partner Payoff (in cents) | −0.0000 (0.0002) | 0.0001 (0.0002) | −0.0007 (0.0004) | 0.0004 (0.0002) | −0.0002 (0.0002) |
| Constant | −0.0900 (0.5139) | 0.7394 (0.5172) | 0.4199 (0.7921) | −0.5909 (0.6317) | −0.3232 (0.5509) |
| Age and Gender | yes | yes | yes | yes | yes |
| Human DM | yes | yes | yes | yes | yes |
| Could choose DM | yes | yes | yes | yes | yes |
| Group Information | yes | yes | yes | yes | yes |
| Direction of Bias | yes | yes | yes | yes | yes |
| Adjusted R-sq | 0.67 | 0.58 | 0.64 | 0.73 | 0.65 |
| Observations | 334 | 140 | 192 | 142 | 334 |

Standard errors clustered at the individual level are in parentheses. * $p<0.05$, ** $p<0.01$, *** $p<0.001$. Note: All dependent variables are the evaluations after learning the monetary value of tokens, ranging from −3 to 3. (2) "Realized = Preferred" reports the estimation for the subsample in which participants' preferred and realized redistributions were identical. (3-4) "Redistributed" and "Not Redistributed" report the estimation for the subsamples in which tokens of any type were redistributed or none of the tokens were redistributed. Controls include age, gender and features of the between-subject variation described in Chugunova and Luhan [33]: whether the redistribution is decided by a human, whether participants could choose who redistributes (DM), whether the minimal group paradigm was used and whether because of it the participant could face a positive, negative or no discrimination in the match based on the group affiliations.

Initially, leaving the token distribution as is was perceived as fairer than redistributing (two-sided t-test, p < 0.001). In columns 3 and 4, we split the sample by whether any of the tokens were redistributed and consider adjustments to fairness separately. We find that in both subsamples participants revise their fairness evaluations upwards when monetary values were revealed and they earned more. Therefore, the self-serving bias is not limited to situations where redistribution occurred but can also be observed when redistribution could have taken place but did not. Interestingly, the latter situation might be even more influenced by the presence of the monetary frame: participants used their preferred redistribution as a reference point and decreased their fairness rankings when the preferred redistribution would yield a higher payoff.

Finally, we turn to the changes in satisfaction ratings. Participants, on average, indicated that they were "somewhat satisfied" with the redistribution (1.06 on the scale between −3 and 3). The OLS regression suggests that the pattern detected for fairness is even more pronounced for satisfaction rankings (Table 1, column 5). Self-serving bias may be more pronounced in the satisfaction measure than in the fairness measure as participants might feel satisfied with the outcome even when they recognize it as unfair. S2 Table suggests that the higher magnitude of coefficients for the satisfaction than for the fairness rankings is persistent for most of the considered subsamples.

## Discussion and conclusion

We examined how knowing the exact monetary value of experimental currency units affects the fairness and satisfaction ratings of a redistribution decision. Importantly, we narrow down the role of self-interest in the evaluation as such and consider the role of *knowing the monetary equivalent* of the redistribution.

We conducted an online experiment where participants earned an income that was then redistributed by a third party. Participants evaluated the same distribution twice, once knowing only token amounts with uncertain monetary value, and once knowing their monetary value.

Learning the monetary values of the tokens significantly impacts the satisfaction and fairness ratings of the same distribution: The ratings in both categories increased with the monetary payoff. These evaluations were adjusted although the token earnings and the relative earning distributions were known before and remained unchanged. As expected, participants who received exactly the distribution they classified as fair ex-ante are least affected by knowing the values of the tokens, and the effect is mainly driven by people who received different distributions. The effect of the monetary value of tokens persists, regardless of whether any tokens were actually redistributed. We find that people adjusted their satisfaction ratings more than their fairness ratings, suggesting the presence of the normative component in the fairness evaluations.

Given our experimental design, participants were given two points of information before making their final evaluation: the value of the tokens and the pair that was selected for payment. We base our conclusions on two established theories, self-interest [41] and motivated reasoning (self-serving bias) [10]. One could argue that learning the payoff-relevant *pair* might also affect the evaluation, but to the best of our knowledge, there is no theoretical background that would support this argument. The literature that appears the closest considers paying one experimental decision vs. paying all, and Charness et al. [18] found that paying one out of several periods does not alter behavior significantly.

Our results contribute to the literature on stake sizes and hypothetical play in economic experiments. While our participants know exactly how many tokens they and their counterparts will earn, they only know a possible range of monetary values for the tokens. This does not confuse them, as their evaluation of the situation stated in terms of tokens and in terms of money is highly correlated. Our results are in line with motivated reasoning and self-serving biases, where subjects follow (or at least state) different principles when the monetary value is known and when it is uncertain. For research designs this means that a salient exchange rate of experimental currency to real money is necessary, at least in games involving social preferences and fairness concerns, to increase the ecological validity of reported results.

## Supporting information

**S1 Table. Ordered Logit regression: Adjustment of fairness and satisfaction rankings after learning the token value.** *Note:* All dependent variables are the evaluations after learning the monetary value of tokens, ranging from –3 to 3. (2-3) "Realized = Preferred" and "Realized ≠ Preferred" reports the estimations for the subsamples in which participants' preferred and realized redistributions were identical or not, respectively. (4-5) "Redistributed" and "Not Redistributed" report the estimation for the subsamples in which tokens of any type were redistributed or in which none of the tokens were redistributed. Controls include age, gender and features of the between-subject variation described in [33]: whether the redistribution is decided by a human, whether participants could choose who redistributes (DM), whether minimal group paradigm was used and whether the participant would face a positive, negative or no discrimination in the match based on the group affiliations.
(PDF)

**S2 Table. OLS regression: Adjustment of satisfaction rankings after learning the token value.** *Note:* All dependent variables are the evaluations after learning the monetary value of tokens, ranging from –3 to 3. (2-3) "Realized = Preferred" and "Realized ≠ Preferred" report the estimations for the subsamples in which participants' preferred and realized redistributions were identical or not, respectively. (4-5) "Redistributed" and "Not Redistributed" report the estimation for

the subsamples in which tokens of any type were redistributed or in which none of the tokens were redistributed. Controls include age, gender and features of the between-subject variation described in [33]: whether the redistribution is decided by a human, whether participants could choose the decision maker (DM), whether minimal group paradigm was used and whether the participant would face a positive, negative or no discrimination in the match based on the group affiliations. (PDF)

**S1 Fig. Snippet of the decision screen.** Initial evaluations. *Note:* This is an example of the screens as they were presented to participants to elicit their initial evaluations of fairness and satisfaction with the redistribution. The layout is an exact representation, the text is a translation as experiment was conducted in German. Complete instructions can be found at OSF: https://osf.io/jucev/.
(PDF)

**S2 Fig. Snippet of the decision screen.** Adjusted evaluations. *Note:* This is an example of the screens as they were presented to participants to elicit their adjusted evaluations of fairness and satisfaction with the redistribution. The layout is an exact representation, the text is a translation as experiment was conducted in German. Complete instructions can be found at OSF: https://osf.io/jucev/.
(PDF)

## Acknowledgments

We thank Nishan Lin and Nisa Erarslan for their assistance

## Author contributions

**Conceptualization:** Marina Chugunova, Wolfgang J. Luhan.

**Data curation:** Marina Chugunova, Wolfgang J. Luhan.

**Formal analysis:** Marina Chugunova, Wolfgang J. Luhan.

**Funding acquisition:** Marina Chugunova, Wolfgang J. Luhan.

**Investigation:** Marina Chugunova, Wolfgang J. Luhan.

**Methodology:** Marina Chugunova, Wolfgang J. Luhan.

**Project administration:** Marina Chugunova, Wolfgang J. Luhan.

**Resources:** Marina Chugunova, Wolfgang J. Luhan.

**Software:** Marina Chugunova, Wolfgang J. Luhan.

**Validation:** Marina Chugunova, Wolfgang J. Luhan.

**Visualization:** Marina Chugunova, Wolfgang J. Luhan.

**Writing – original draft:** Marina Chugunova, Wolfgang J. Luhan.

**Writing – review & editing:** Marina Chugunova, Wolfgang J. Luhan.

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
