## [Decision Letter · Decision Letter 0]

17 Nov 2025

PONE-D-25-55896

A Matter of Principle or a Matter of Money? How Fairness Evaluations Change with Experimental Currencies.

PLOS ONE

Dear Dr. Luhan,

Thank you for submitting your manuscript to PLOS ONE. After careful consideration, we feel that it has merit but does not fully meet PLOS ONE’s publication criteria as it currently stands. Therefore, we invite you to submit a revised version of the manuscript that addresses the points raised during the review process.

**ACADEMIC EDITOR:** One reviewer has suggested the rejection of the paper, given flaws in your experimental design. However, i spent some time with your manuscript and as the concern and research question is valid. I would like to request you to improve your manuscript.

We look forward to receiving your revised manuscript.

Kind regards,

Raja Rajendra Timilsina, PhD

Academic Editor

PLOS ONE

Journal Requirements:

The authors gratefully acknowledge funding by the Nuffield Foundation (FR-000000326) and DFG CRC TRR 190 (project number 280092119).

3. Please expand the acronym “DFG CRC TRR” (as indicated in your financial disclosure) so that it states the name of your funders in full.

5. We noted in your submission details that a portion of your manuscript may have been presented or published elsewhere. Please clarify whether this publication was peer-reviewed and formally published. If this work was previously peer-reviewed and published, in the cover letter please provide the reason that this work does not constitute dual publication and should be included in the current manuscript.

6. Please update your submission to use the PLOS LaTeX template. The template and more information on our requirements for LaTeX submissions can be found at http://journals.plos.org/plosone/s/latex.

Reviewers' comments:

Reviewer's Responses to Questions

**Comments to the Author**

1. Is the manuscript technically sound, and do the data support the conclusions?

Reviewer #1: Yes

Reviewer #2: No

2. Has the statistical analysis been performed appropriately and rigorously?

Reviewer #1: Yes

Reviewer #2: Yes

3. Have the authors made all data underlying the findings in their manuscript fully available?

Reviewer #1: Yes

Reviewer #2: Yes

4. Is the manuscript presented in an intelligible fashion and written in standard English?

Reviewer #1: Yes

Reviewer #2: Yes

5. Review Comments to the Author

Reviewer #1: [General]

This paper addresses a highly relevant and timely question in experimental economics. The topic is conceptually important and methodologically elegant, contributing meaningfully to the literature on fairness perception and self-serving bias. The experimental design is clear and replicable, and the interpretation of results is balanced.

→ Recommendation: Revision

[Minor Comments]

Overall, I recommend only minor revisions.

Clarify the rationale for task design (luck–effort–talent).

On page 6 (lines 138–141), the manuscript states:

“Participants could earn 0 or 100 tokens via a coin toss (luck); they counted zeros in two matrices (effort); finally, they could earn 100 tokens for solving one Raven’s Progressive Matrix (talent).”

These tasks are appropriate and intuitive, but the paper would benefit from a slightly fuller justification grounded in prior experimental literature. In particular, a short paragraph explaining why these tasks are suitable operationalizations of luck, effort, and talent would strengthen the methodological rationale.

In sum, this is a well-executed and thought-provoking paper that makes a valuable methodological and conceptual contribution. With a minor clarification of the task rationale and a few stylistic improvements, it will be fully ready for publication.

Reviewer #2: Reviewer report

A Matter of Principle or a Matter of Money? How Fairness Evaluations Change with Experimental Currencies

Authors: Marina Chugunova, Wolfgang J. Luhan

Review Date: 16/11/2025

1. Overall assessment and major critique

The manuscript addresses a potentially important methodological question regarding how monetary framing influences fairness evaluations in economic experiments. While the topic is relevant and the experimental approach shows some thoughtful elements, a fundamental methodological confound severely undermines the validity of the paper's central conclusions.

The authors claim to demonstrate that "knowledge of monetary value can serve as a frame" that intensifies self-serving bias in fairness evaluations. However, the experimental design fails to isolate the monetary framing effect from another critical variable: the transition from evaluating multiple hypothetical scenarios to assessing a single, concrete, payoff-relevant outcome. This confound represents a fatal flaw in the study's internal validity.

2. Detailed methodological critique

2.1 The Critical Confound: Bundled Treatments

The experimental procedure simultaneously introduces two distinct psychological treatments:

• TREATMENT A (Monetary Framing): Conversion of abstract tokens to concrete monetary values (4¢, 2¢, 3¢)

• TREATMENT B (Outcome Concretization): Identification of one specific pair as the actual determinant of final payment

The current design administers both treatments concurrently before the final evaluation. The observed shift in fairness ratings could therefore be attributed to:

• Treatment A alone (as the authors claim)

• Treatment B alone (the act of making an outcome real)

• An interaction between A and B

The literature strongly suggests that Treatment B—moving from hypothetical to real stakes—is itself a powerful driver of self-interested evaluation. The personal salience of knowing "this specific outcome determines my payment" likely triggers emotional engagement and self-serving bias independent of monetary quantification.

2.2 Required Design Modification

To properly test the authors' hypothesis, the experimental procedure should have isolated the monetary framing effect. A methodologically sound approach would be:

1. Reveal the payoff-relevant pair immediately after the earning phase

2. Collect initial fairness evaluations for this known payoff-relevant pair (in tokens)

3. Reveal monetary values and collect final evaluations

This design would complete the abstract-to-concrete transition before the first measurement, allowing any subsequent rating changes to be more confidently attributed to the monetary frame itself.

3. Implications for interpretation

Given this confound, the authors cannot validly claim that "knowledge of monetary value" causes the observed shift in fairness evaluations. The alternative explanation—that outcome concretization drives the effect—is equally plausible based on the presented data.

The paper's central contribution ("demonstrating that knowledge of monetary value can serve as a frame") therefore remains unsubstantiated. The results may instead demonstrate that making outcomes real (rather than specifically monetary) triggers self-serving evaluations.

4. Additional editorial notes

The manuscript contains several minor grammatical and phrasing issues that require attention, including inconsistent hyphenation in "crowding-out," missing possessive apostrophes in references to theoretical concepts, subject-verb agreement errors, and inconsistent use of terminology between "satisfaction" and "happiness." While these do not affect the scientific argument, they detract from the manuscript's polish.

5. Final recommendation

The identified methodological confound is fundamental to the experimental design and cannot be addressed through reanalysis or revision. The study fails to provide compelling evidence for its central thesis due to this validity threat. I therefore recommend rejection.

The research question remains valuable, and I would encourage the authors to address this design issue in future work exploring the relationship between monetary framing and fairness evaluations.

6. PLOS authors have the option to publish the peer review history of their article (what does this mean?). If published, this will include your full peer review and any attached files.

Reviewer #1: No

Reviewer #2: No

---

## [Author Response · Author response to Decision Letter 1]

5 Dec 2025

We have addressed all comments - as requested - in the attached response letter.

Unfortunately the system does not allow us to order the files, so the response letter can be found after the manuscript with tracked changes.

---

## [Editor Report · Decision Letter 1]

16 Dec 2025

A Matter of Principle or a Matter of Money? How Fairness Evaluations Change with Experimental Currencies.

PONE-D-25-55896R1

Dear Dr.Wolfgang Luhan,

We’re pleased to inform you that your manuscript has been judged scientifically suitable for publication and will be formally accepted for publication once it meets all outstanding technical requirements.

Kind regards,

Raja Rajendra Timilsina, PhD

Academic Editor

PLOS One
---

## [Editor Report · Acceptance letter]

PONE-D-25-55896R1

PLOS One

Dear Dr. Luhan,

I'm pleased to inform you that your manuscript has been deemed suitable for publication in PLOS One. Congratulations! Your manuscript is now being handed over to our production team.

Kind regards,

on behalf of

Dr. Raja Rajendra Timilsina

Academic Editor

PLOS One